# Knowledge Distillation based Robot-Object Manipulation Failure Anticipation

Tugce TEMEL, Arda INCEOGLU and Sanem SARIEL

*Abstract*— An autonomous service robot should be capable of interacting with its environment safely. However, outcomes of executions are not always as expected due to several factors including perception errors, failures in manipulation, or unexpected external events. While most current research emphasizes detecting and classifying robot failures, our study shifts its focus to anticipating these failures before they occur. The underlying idea is that by anticipating a potential failure early on, preventive actions can be taken. To address this, we present a novel failure anticipation framework based on knowledge distillation. This system utilizes video transformers and incorporates a sensor fusion network designed to handle RGB, depth, and optical flow data. We assess the effectiveness of our method on FAILURE, a real-world robot manipulation dataset. Experimental results indicate that our proposed framework achieves an F1 score of 82.12%, highlighting its ability to anticipate robot execution failures up to one second in advance.

## I. INTRODUCTION

Robots interact with humans and objects in various social, unstructured environments. In such environments, robots need a variety of skills to ensure safe interaction [1]. During these interactions, unforeseen situations can arise due to incorrect estimation of parameters or representation of the real world. Such situations raise concerns for safety and security in robot-object interaction.

In order to enhance safety during interactions between robots and objects, continuous execution monitoring is necessary. For this purpose, there have been studies for failure detection [2], [3], [4], [5], [6], [7] and classification [3], [8], [9]. In these works, a post-execution classification is targeted indicating whether the execution was failed or successful. Nevertheless, this is only helpful as an inspection procedure after a failure occurs. In contrast, failure anticipation involves determining whether there will be a failure before its actual occurrence. Hence, anticipating and preventing failures is possible. This proactive strategy aims to reduce the potential harm that may arise from failures in manipulation. Fig. 1 presents the difference of a failure anticipation case from a failure detection case on a pouring scenario.

In a recent study, a robot was trained to learn failure-preventive actions using deep reinforcement learning [10]. In this work, rule-based methods are utilized to predict a potential risk that may pose a threat to safety. The method

This research is supported by a grant from the Scientific and Technological Research Council of Turkey (TUBITAK), Grant No. 119E-436.

All authors are with the Artificial Intelligence and Robotics Laboratory, Faculty of Computer and Informatics Engineering, Istanbul Technical University, Maslak, Turkey
{temel21, inceoglua, sariel}@itu.edu.tr

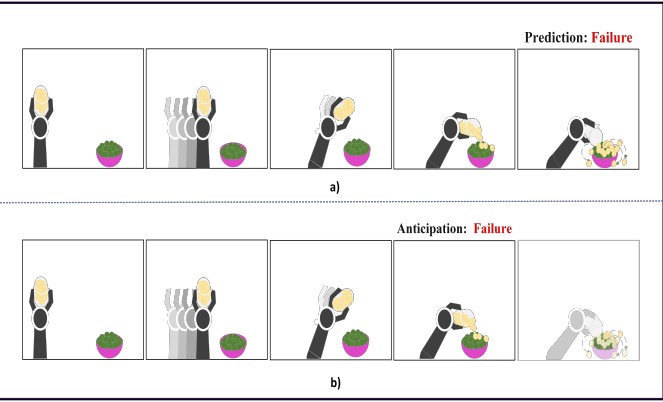

Fig. 1: Symbolic representations for (a) detecting and (b) anticipating a failure during a scenario involving a robot pouring into a bowl. The input of the detection problem contains the frames including the failure case. Yet these frames are excluded for the anticipation problem, and the goal is to predict the failure.

we propose aims to predict risks without expert knowledge and predefined features, using only real-world data.

Early detection refers to the earliest possible identification of an event, while anticipation involves predicting an event before it occurs [11]. Similar to the problem of failure anticipation, there have been studies on other domains such as human activity anticipation [12]. For instance, [13] uses exponential loss for delayed predictions, while another study [14] combines cross-entropy loss with ranking loss. In another work [15], the Rolling-Unrolling LSTM method is being utilized. In this method, Rolling LSTM encodes historical context into a hidden vector, and Unrolling LSTM uses this hidden vector for decoding.

Knowledge distillation is the process of transferring knowledge from a large, complex model (Teacher model) to a smaller, lighter model (Student model) to adapt it to real-time applications [16], [17], [18]. In contrast, in this work, the student network is designed to have the identical capacity as the teacher network. Knowledge distillation has also been used in predicting human activities [19], [20]. In this study, knowledge distillation is reinterpreted for anticipating robot-object manipulation failures with a multi-headed transformer backbone network (Section II-C).

Spatio-temporal sensory data analysis is necessary to represent the real-world robot manipulations which are used for failure anticipation. In recent years, video transformers have shown significant advancements in computer vision by utilizing self-attention mechanisms to effectively capture complex interactions, making them ideal for tasks like video

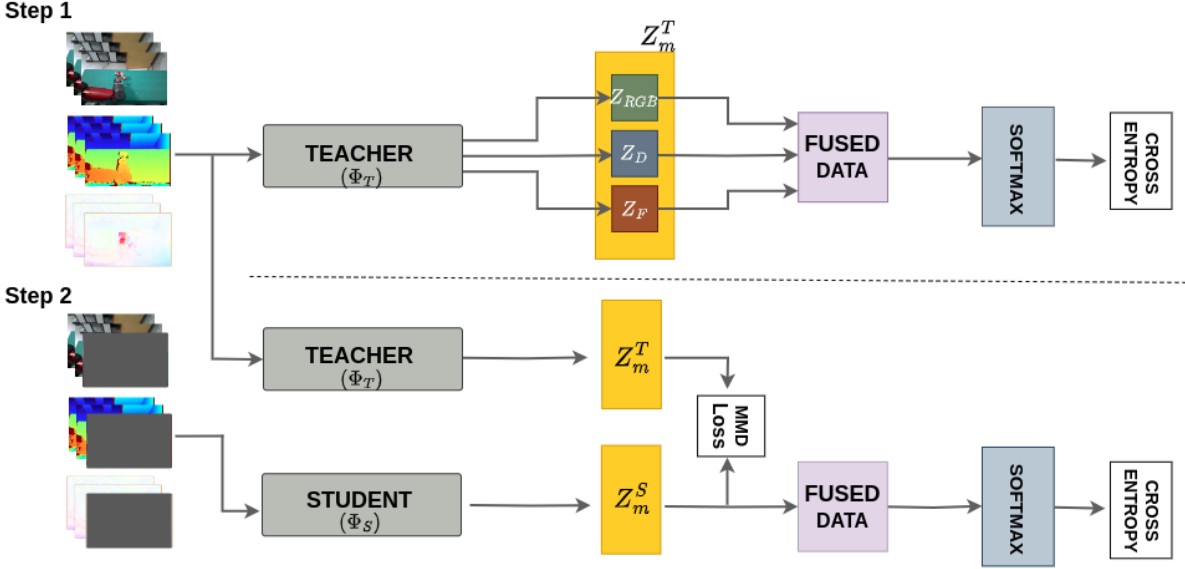

Fig. 2: Knowledge distillation based failure anticipation architecture. Training has two steps. First, the *Teacher* network is trained independently of the *Student*. Then, the *Teacher* network weights are frozen and used to guide the *Student* network's training by distilling learned representations.

processing. Therefore, ViViT [21] is used as the backbone model for our study.

In summary, the contributions of this study are as follows:

- we introduce extended FAILURE dataset [2] with optical flow modality,
- we present a sensor fusion based transformer model for RGB, depth and optical flow modalities,
- we propose a failure anticipation architecture that uses knowledge distillation.

## II. METHODOLOGY

In this study, a knowledge distillation-based failure anticipation architecture is presented (Fig. 2). The proposed architecture consists of *Teacher* ($\Phi_T$) and *Student* ($\Phi_S$) networks. While the *Teacher* network assists in the learning process, failure anticipation is performed by the *Student* network. Both the *Teacher* and *Student* networks are implemented based on video transformers models.

### A. Problem Description

Anticipating manipulation failures requires carefully observing changes in the scene using multiple sensory modalities [2], [3]. In this study, the failure anticipation task is represented as a classification problem without the failure information (i.e., input does not contain frames corresponding to the failure). Consider a set of sensing modalities denoted as $M \in \{1,2,3\ldots\}$ where $m \in M$ represents the modality index. Let $D$ be the dataset containing $|D| = N$ multimodal observation sequences. An observation is denoted by $x_{t_m,i}^m$, $t_m$ represents the time index for modality $m$, and $i$ is the recording index. Finally, $y \in \{fail, success\}$ is the label:

$$D = \{\{(x_{1,i}^m \ldots x_{t_m,i}^m)\}_{m=1}^M, y_i\}\}_{i=1}^N \quad (1)$$

The objective is to anticipate failures by extracting discriminative features from $\Phi_T(.)$, which is specified for failure detection. Subsequently, these features are distilled to $\Phi_S(.)$ which assigns (i.e., anticipation) a categorical label as either success or failure to the subset of multimodal sensory data.

### B. Dataset and Preprocessing

In this study, the FAILURE dataset [3] is utilized which consists of 324 real-world object interaction video recordings captured using the Baxter robot, containing RGB, depth, and audio modalities. These object interactions are categorized as *pouring, pushing, place-in-container, put-on-top*, and *pick&place*. A symbolized *pouring* execution failure is given in Fig. 1. Additionally, we extended the FAILURE dataset with optical flow modality. Optical flow represents a distribution that expresses the apparent velocities of brightness patterns' movement in an image. This movement can originate from the moving entity or passively affected objects relative to it [22]. Therefore, optical flow provides significant information about the spatial arrangement of objects causing or being affected by the motion [23].

Positions, distances, and motion states of objects in the

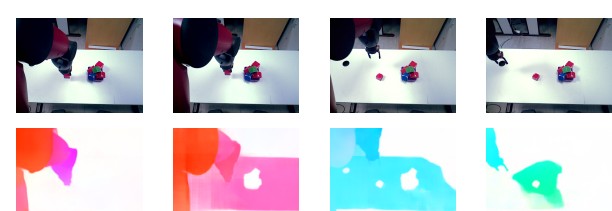

Fig. 3: RGB frames captured using the head camera during the robot-objects interaction and their corresponding optical flow frames.

environment are crucial for identifying the signatures of failures. Therefore, we propose to use depth and optical flow data in addition to RGB data. The optical flow data obtained from RGB using FlowNet2 [24] is merged into the dataset (Fig. 3).

**Data Preprocessing.** Due to the online planning and execution of object interaction trajectories by the robot, the length of recordings varies depending on the type of object interaction and the planned trajectory. Each recording in the dataset is downsampled to 1 FPS, converting them into sequences of sequential image frames.

Videos in the dataset are represented by sampled 8 frames including the failure. While the *Teacher* network can observe all 8 frames, the selection for the *Student* network is constrained to be just before the occurrence of a failure. Student network observes 7 frames from the execution and a blank frame. Finally, all frames are cropped to 224 × 224 pixels corresponding to the dimensions of the table.

### C. Network Architecture

The proposed model has been designed to anticipate potential failures and provide enough time to prevent them before they occur. The key aspect of the model is the use of the knowledge distillation method to anticipate failures. Knowledge distillation architecture is composed of *Teacher* and *Student* networks. Both networks utilize Video Vision Transformer model (ViViT [21]) as backbone.

**Teacher Network:** The teacher network is responsible for failure detection. It has access to all frames of the manipulation execution. The teacher model ($\Phi_T(.)$) is formulated as follows:

$$\hat{y}_{t_m,i} = \Phi_T(\phi_1(x_{1,1}^{(1)}, \ldots, x_{t_1,i}^{(1)}) \oplus \cdots \oplus \phi_m(x_{1,1}^{(m)}, \ldots, x_{t_m,i}^{(m)})) \quad (2)$$

**Student Network.** The main goal of the student network is to anticipate potential failures in advance. The input of the Student network is denoted by $x_{r_m,i}^m$. $r$ contains a subset of time indices that correspond to the frames before the failure which is defined as $r \in \{1, 2, \ldots, (t-1)_m\}_{m=1}^M$. The *Student* ($\Phi_S$) is formulated as:

$$\hat{y}_{t_m,i} = \Phi_S(\phi_1(x_{1,1}^1, \ldots, x^{(1)}r_1, i) \oplus \cdots \oplus \phi_m(x_{1,1}^{(m)}, \ldots, x^{(m)}r_m, i)) \quad (3)$$

**Vision Transformer:** The ViViT [21] model-3 architecture is adopted to extract features from failure manipulation videos. We input a sequence of 8 video frames with 224 x 224 pixels dimensions into ViViT. The size of the extracted tubelet from each image frame is defined by ($patch_h$, $patch_w$) = 32 x 32. Additionally, the temporal tubelet size ($patch_t$) is set to 4. The dimension parameter is specified as 256, while the depth parameter is set to 6. Moreover, a multi-head self-attention mechanism with a head size of 8 is utilized.

### D. Training Methodology

In this work, offline distillation is adopted [25], [26], [27] where training involves two steps. First, the teacher is trained independently of the student. Then, the teacher network weights are frozen and used to guide the student network's training by distilling learned representations.

*Teacher* network is trained using cross-entropy loss. On the other hand, the loss function for the *Student* network is composed of two parts. One of which measures the loss of the student network's failure anticipation ($L_C$) and the other one is used to measure the distillation loss ($L_{TS}$) from the teacher to the student network. The total loss ($L_{total}$) function is formulated as:

$$L_{total} = \alpha L_{TS}(Z_m^T, Z_m^S) + \beta L_C(y, \hat{y}) \quad (4)$$

Here, $\alpha$ and $\beta$ are defined as learnable parameters of the network, and their values are determined by the network during training. $L_C$ represents the failure anticipation loss of the student network and is calculated using cross-entropy (CE). $y$ denotes the true class label, and $\hat{y}$ represents the student's anticipated class label. The distillation loss, denoted as $L_{TS}$, is calculated using the maximum mean discrepancy (MMD) loss function and formulated as:

$$L_{TS} = L_{MMD}(Z_{RGB}^T, Z_{RGB}^S) + L_{MMD}(Z_D^T, Z_D^S) + L_{MMD}(Z_F^T, Z_F^S) \quad (5)$$

### III. EXPERIMENTS

For quantitative evaluation, the extended FAILURE dataset is divided into training (70%), validation (10%), and test sets (20%). All network weights are initialized randomly, and training is conducted for 250 steps using Adam optimization, with a learning rate set to 1e-5. The best model is determined based on the validation set using an early stopping strategy. Test scores obtained with the selected best models are reported in the following sections. Data augmentation is applied to prevent overfitting, where, for example, the brightness, contrast, saturation, and hue values of all images in a sequence are randomly altered with a 20% probability. Similarly, each image sequence is flipped vertically with a 50% probability.

### A. Quantitative Evaluation

In this section, the numerical results obtained are presented. The proposed architecture has been trained for both individual modalities and multi-modal sensor fusion. In the multi-modal architecture, an independent copy of the transformer network is created for each modality, and the modalities are combined through late fusion.

TABLE I: QUANTITATIVE RESULTS FOR TEACHER NETWORK

|  | Precision | Recall | F1 Score |
| --- | --- | --- | --- |
| **RGB** | 62.63 | 62.16 | 62.39 |
| **D (Depth)** | 64.97 | 60.89 | 62.86 |
| **F (Optical Flow)** | 64.57 | 62.16 | 63.34 |
| **RGB-F** | 67.56 | 67.56 | 67.56 |
| **RGB-D** | 70.43 | 67.56 | 68.97 |
| **RGB-D-F** | **76.05** | **75.67** | **75.86** |

Table I presents the results obtained from the *Teacher* network for different sensor modalities and their fusion. The input frames of the *Teacher* network cover the entire video. Therefore, it represents a failure detection problem.

The results show that the multimodal network outperforms one with a single modality for failure detection. Multimodal fusion is significant in enhancing the accuracy and reliability of failure detection systems by leveraging the complementary strengths of each modality. Combining all modalities (RGB-D-F) yields the highest performance with an F1 score of 76.06% during testing.

TABLE II: Quantitative Results for Student Network (Failure Anticipation)

|  | 1-Frame (1 sec) | | | 2-Frames (2 sec) | | |
|---|---|---|---|---|---|---|
|  | Precision | Recall | F1 Score | Precision | Recall | F1 Score |
| **RGB** | 73.05 | 72.97 | 72.98 | 72.64 | 62.16 | 66.99 |
| **D** | 68.26 | 68.42 | 68.34 | 66.84 | 65.78 | 66.31 |
| **RGB-D** | 75.85 | 75.67 | 75.76 | 70.19 | 70.27 | 70.23 |
| **RGB-F** | 79.41 | 78.37 | 78.89 | 67.48 | 67.52 | 67.52 |
| **RGB-D-F** | **83.19** | **81.08** | **82.12** | **81.27** | **78.37** | **79.79** |

Table II presents the results of the *Student* network. The input to the *Student* network includes frames preceding the failure and is utilized for failure anticipation. The column *1-Frame* in the table represents the last frame observed by the student network before the failure (i.e., 1 second before the failure). When comparing the performance of the student network for the *1-Frame* column with that of the teacher network, the student network outperforms the teacher network for each modality.

The student network is designed to have the identical capacity as the teacher network, contrary to conventional approaches [28], [16], [17], [18]. Additionally, despite not directly observing failures, the student network distils all the information possessed by the teacher network for failure anticipation and further validates it by comparing the learned information with the ground truth data labels. Hence, it exhibits better performance than the teacher network for the *1-Frame* column. When comparing *2-Frames* with *1-Frame*, since *1-Frame* is closer to the failure event, the failure anticipation F1 score of the student network is higher as expected.

## IV. CONCLUSION

In summary, anticipating failures is a crucial step in ensuring the reliability and robustness of autonomous systems. Real-time execution monitoring approaches for object manipulation can detect potential failures, and the relevant preventive actions can be taken before undesired outcomes occur. Our proposed method enables the robot to anticipate object manipulation failures with an F1-score of 82.12% 1 second in advance and 79.79% 2 seconds in advance. The obtained results demonstrate the potential utility of our proposed system in preemptively anticipating failures. Real-time testing of the developed video transformer-based architecture on the robot is an ongoing work.

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
