# OpenReview forum: "Knowledge Distillation based Robot-Object Manipulation Failure Anticipation"
_IEEE.org/2024/ICRA/Workshop/CookingRobot — CookingRobot2024 Poster_

### Official Review · Reviewer_mnet · 2024-04-10
**The review for Knowledge Distillation based Robot-Object Manipulation Failure Anticipation**

**Rating:** 7
**Confidence:** 5

**Review:**

*Major Contribution of the Paper

The paper proposes a failure anticipation framework, utilizing knowledge distillation, where a Teacher network trained on a multi-modal dataset is used to guide the learning of a Student network.
The paper would contribute to the field of cooking robotics, specifically in enhancing safety and reliability during object manipulation tasks.

*Major comments:

I felt that detecting whether a failure will occur in the next frame using a sequence of 7 frames may be too short of a span. What is the time interval between frames? If it's too short, isn't it possible that the object is already in a failed state within the 7 frames being used?

*Video

The video shows a good overall summary of the research. I would like to see the anticipation result with videos on the fly.

*Paper

The paper and the video are well-structured, providing clear explanations of problem formulations and network architectures.

*Minor comments

Reference [9] is missing the author names.

---

### Official Review · Reviewer_oLGT · 2024-04-16
**The review for ”Knowledge Distillation based Robot-Object Manipulation Failure Anticipation”**

**Rating:** 7
**Confidence:** 3

**Review:**

This paper proposes a framework based on knowledge distillation to predict robot failures in advance from RGB, depth, and optical flow data. Cooking is a complex and failure-prone task, so this paper's proposal should be an important topic of discussion.

Major Comments:
* Table II shows that the accuracy at 2-Frames (2sec) is around 80%, but it is hard to believe that it is possible to predict failures from 2 seconds old data in the case of human manipulation. The robot manipulation data is probably moving at a much slower speed than humans, but it is interesting to know what kind of acutal input is being used to make predictions, and whether the robot is learning different features from the failure data rather than predicting failures.

Video:
* This is a good explanatory video to help understand the paper. Material that allows discussion of qualitative evaluation, such as showing examples of successes and failures, would be helpful for better discussions in workshops.